# Crafting Good Views of Medical Images for Contrastive Learning via Expert-level Visual Attention

**Sheng Wang**                                           WSHENG@SJTU.EDU.CN
**Zihao Zhao**                                        ZIHAOZHAO10@GMAIL.COM
**Lichi Zhang**                                       LICHIZHANG@SJTU.EDU.CN
**Dinggang Shen**                               DGSHEN@SHANGHAITECH.EDU.CN
**Qian Wang**                                WANGQIAN2@SHANGHAITECH.EDU.CN

**Editor:** Editor's name

## Abstract

Recent advancements in contrastive learning methods have shown significant improvements, which focus on minimizing the distances between different views of the same image. These methods typically craft two randomly augmented views of the same image as a positive pair, expecting the model to capture the inherent representation of the image. However, random data augmentation might not fully preserve image semantic information and can lead to a decline in the quality of the augmented views, thereby affecting the effectiveness of contrastive learning. This issue is particularly pronounced in the domain of medical images, where lesion areas can be subtle and are susceptible to distortion or removal. To address this issue, we leverage insights from radiologists' expertise in diagnosing medical images and propose Gaze-Conditioned Augmentation (GCA) to craft high-quality contrastive views of medical images given the radiologist's visual attention. Specifically, we track the gaze movements of radiologists and model their visual attention when reading to diagnose X-ray images. The learned model can predict visual attention of the radiologist when presented with a new X-ray image, and further guide the attention-aware augmentation, ensuring that it pays special attention to preserving disease-related abnormalities. Our proposed GCA can significantly improve the performance of contrastive learning methods on knee X-ray images, revealing its potential in medical applications.

**Keywords:** Eye-tracking, Contrastive Learning, Medical Image, Human Visual Attention.

## 1. Introduction

Contrastive learning has made remarkable advancements, even matching the performance of supervised pre-training in various downstream tasks (Chen et al., 2020a; He et al., 2020; Grill et al., 2020). At its core, the essence of contrastive learning lies in the presentation of diverse, transformed views of a single image. The underlying expectation is for the model to grasp the intrinsic characteristics embedded within this image. These diverse views, often characterized as positive pairs, are intended to be closely aligned in the latent space, while disparate views from distinct samples form negative pairs.

The creation of high-quality positive pairs is a pivotal focus in contrastive learning, as highlighted in recent works (Grill et al., 2020; Chen and He, 2021; Caron et al., 2021). On natural images, a common approach to generating positive pairs involves applying random augmentations twice to a specific image, such as random cropping, color distortion, rotation, cutout, noise addition, and more (Zhong et al., 2020; Chen et al., 2020b).

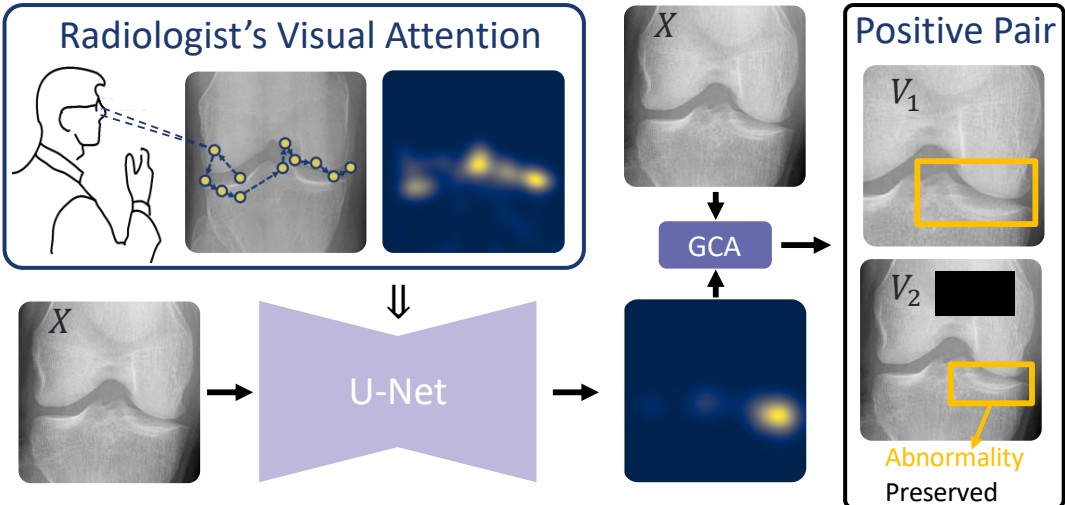

**Figure 1:** Overview of our proposed framework. A saliency prediction model is adopted to mimic expert-level visual attention from the radiologist, and the predicted gaze heatmap is utilized to condition the generation of positive pairs through our proposed GCA. Through this process, we ensure that critical information, such as abnormal areas, is effectively preserved in the resulting contrastive views.

However, this process often fails to consider image semantics, potentially missing regions of interest crucial for subsequent tasks, which can be pronounced on medical images. For instance, in Figure 1, the yellow-box-highlighted abnormality related to knee osteoarthritis (OA) diagnosis may easily be overlooked when using conventional random cropping or cutout techniques.

Although there have been discussions regarding the optimal design of augmentation strategies for contrastive learning (Tian et al., 2020; Xiao et al., 2020; Dangovski et al., 2021), it remains imperative to exercise cautiously when applying widely accepted techniques, such as random resized cropping, in the domain of medical imaging. As shown in Figure 2, the visual cues related to the diseased abnormality in the X-ray image may only occupy a mere fraction of the total pixel count (e.g., 4.12% of the image area for the abnormality of femur-tibia space narrowing). Nevertheless, these limited pixels are expected to provide pivotal information for facilitating clinical decision-making and diagnosis. These pivotal but few pixels are likely to be removed especially when the augmentation strength becomes excessive, and the resulting positive pairs may yield plain contrastive representations that hardly capture underlying clinical semantics, since the visual cues for subsequent diagnosis are missing. On the contrary, a natural image depicting a dog (as depicted in Figure 2, with 35.50% foreground pixels) is less susceptible to the distortion induced by image augmentation. Thus, given similar image augmentation techniques (as depicted in the rightmost section of Figure 2), the ability to distinguish the image as representing a dog, as opposed to lesion areas, remains largely unaffected.

In this study, we argue that optimal contrastive learning on medical images should follow the human visual system, i.e., not to lose the salient parts in the images that provide critical visual cues to clinical diagnoses. To introduce expert-level visual attention, we first train a

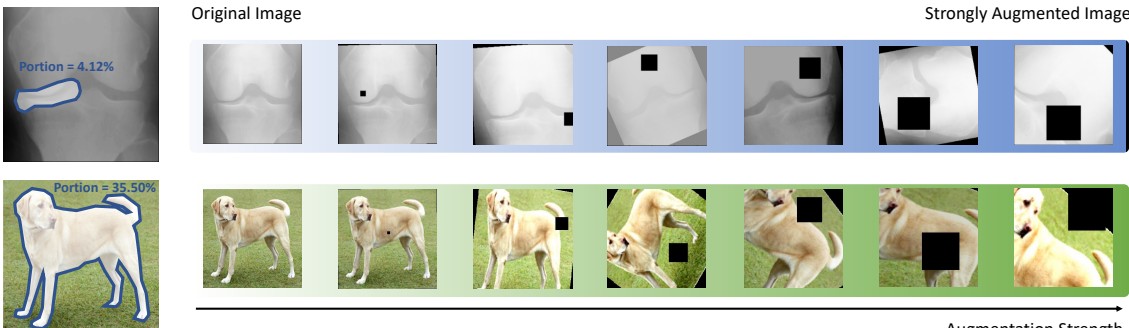

**Figure 2:** Improper contrastive views in medical images due to too strong augmentation. When knee X-rays are heavily augmented, preserving and recognizing semantic cues (i.e., abnormalities) becomes difficult, potentially leading to significant deviations in subsequent diagnostic tasks. Differently, the significantly transformed dog image can still convey most semantic cues, facilitating successful classification tasks.

saliency prediction model (e.g. U-Net) supervised by radiologists' gaze. We then propose Gaze Conditioned Augmentation (GCA), a novel semantic-aware augmentation strategy that replaces conventional, manually designed approaches for medical image augmentation. Subsequently, the GCA can utilize predicted gaze maps to identify and filter out improper data augmentations, thus safeguarding the integrity of salient regions of medical images. It can be delineated into two primary facets: (1) we propose *gaze cutout* and *gaze crop* techniques, designed to judiciously excise non-salient regions from the image. This approach allows for the preservation of disease-related anomalies within positive image pairs, even when these anomalies are minute and prone to obliteration during conventional random augmentation procedures. (2) Leveraging insights from radiologists, who possess profound expertise in the expeditious interpretation of medical images, we employ the *gaze mask* to suppress non-salient areas within the images. This strategic application facilitates the network's ability to differentiate between negative image pairs more effectively.

GCA serves as a plug-and-play module, seamlessly integrating with contrastive learning frameworks (He et al., 2020; Chen et al., 2020a; Grill et al., 2020). Empirical results consistently demonstrate the superior performance of our proposed semantic-aware augmentation technique compared to conventional approaches such as random augmentation or the utilization of manually-tailored hyperparameters.

The structure of this paper unfolds as follows: Section 2 provides an overview of the relevant literature in contrastive learning, with a particular emphasis on the concept of positive pairs. Additionally, it discusses previous efforts to incorporate eye-tracking technology in radiology. In Section 3, we introduce GCA, a novel approach that, to the best of our knowledge, stands as one of the earliest attempts to leverage human visual attention to guide the process of contrastive learning. Our code is publicly available at Github.

## 2. Related Work

### 2.1. Contrastive Learning on Medical Images

Contrastive learning has emerged as a leading self-supervised learning approach, demonstrating superior performance on downstream tasks. Pre-trained networks can outperform fully-supervised networks with only the linear layer being tunable (Tomasev et al., 2022). Researchers also employ linear probing to assess the quality of learned representations by evaluating their performance on subsequent tasks while keeping the pre-trained model's features fixed. Large-scale contrastive pre-training with linear probing has gained popularity due to its generalizability across various scenarios and resilience against overfitting (Radford et al., 2021).

In the context of medical image analysis, several efforts have been made. MoCo-CXR (Sowrirajan et al., 2021) is pre-trained on chest X-ray for improved representations and initializations. Zhou et al. (2020) presented C2L, offering pre-trained 2D deep models for radiograph-related tasks using unannotated data. These works have typically employed a conventional augmentation strategy similar to SimCLR (Chen et al., 2020a).

Initially, high-quality representations in contrastive learning necessitated a large number of negative pairs in a batch. SimCLR demonstrated that contrastive learning benefits from larger batch sizes, allowing for more negative pairs. To overcome GPU memory limitations, Momentum Contrast (MoCo) maintained a momentum memory bank of negative representations. However, recent research has shown that the number of negative pairs has a limited impact on representation quality when the learning framework is designed effectively. For instance, BYOL employs two Siamese networks and an additional non-linear transform, enabling contrastive learning with fewer negative samples in a single batch. DINO extends this concept, utilizing a transformer architecture with a small batch size. Recent research increasingly emphasizes the importance of positive pairs in contrastive learning, a focus we also maintain in this paper regarding their role in representation quality.

### 2.2. Eye-tracking in Radiology

Visual attention is a crucial aspect of radiologists' decision-making processes. Carmody et al. (1981) conducted an early eye-tracking study in radiology, revealing that radiologists' gaze strategies impact their ability to detect lung nodules in X-rays. Kundel et al. (2008) gathered eye-tracking data and found that 57% of cancer lesions were located within the first second of viewing. Voisin et al. (2013) identified a strong correlation between radiologists' gaze patterns and diagnostic errors in mammogram interpretation.

Studies on volumetric CT and MRI images also show that specialists react differently to lesions. Bertram et al. (2016) found that specialists exhibit longer fixations and shorter saccades when encountering lesions. Mallett et al. (2014) reported that experienced radiologists have higher polyp identification rates in CT colonography due to multiple pursuits. Stember et al. (2020) used eye-tracking to label brain tumors in MRI scans.

## 3. Learn to Craft Views from Radiologist's Gaze

Figure 2 illustrates that a simplistic transference of contrastive augmentation techniques from natural images to knee X-ray images is not appropriate. In addressing this challenge,

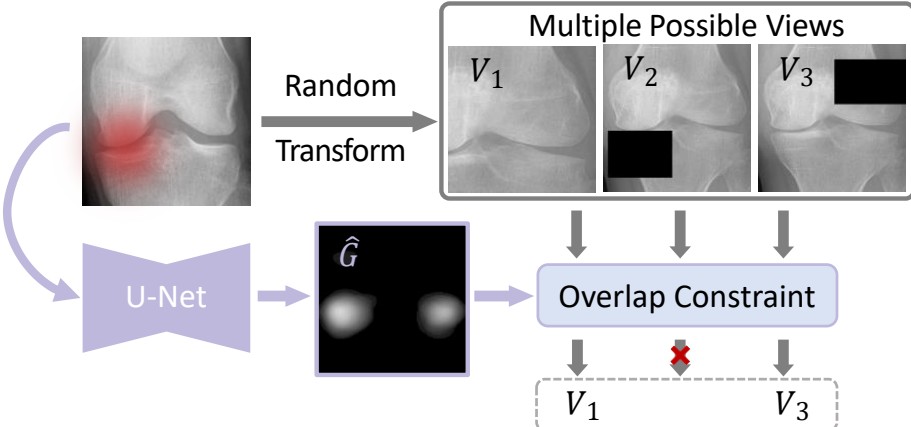

**Figure 3:** The overview of Gaze Conditioned augmentation. A U-Net is trained to predict the radiologist's gaze map. When an image is augmented into multiple views, the predicted gaze map is utilized to decide whether the salient regions in the images are preserved or not. If the salient part is preserved, the augmented view will be preserved for further process in contrastive learning; otherwise, it will be treated as a false-positive view and discarded.

we present a straightforward yet effective approach aimed at acquiring semantic-aware augmentations based on the visual attention patterns of radiologists when examining X-rays. This methodology is depicted in Figure 3. Initially, we collect hundreds of radiologists' gaze data and utilize this data to train a U-Net (Ronneberger et al., 2015) as the gaze map predictor. Subsequently, the trained gaze map predictor plays a pivotal role in evaluating each augmented view, determining its eligibility for integration into the contrastive learning process. In particular, the gaze map predictor serves as a guide, indicating the regions within the (augmented) image that harbor significant semantic information, as per the observed gaze patterns of radiologists. If the augmented view successfully preserves salient elements, it is incorporated into the contrastive learning pipeline; otherwise, it is excluded from consideration.

### 3.1. Gaze Collection and Post-processing

The eye-tracking data for this study was collected using a Tobii 4C remote eye-tracker, which recorded binocular gaze data at a rate of 90Hz. Customized data collection software was developed in Python, utilizing the manufacturer-provided SDK. We are pleased to make this software publicly available in conjunction with this paper. Participants in the study were positioned in front of a 27-inch LCD screen, replicating a clinical working environment, and were given the flexibility to adjust their seating distance from the screen to ensure comfort. Our software meticulously logged the gaze locations on the screen with corresponding timestamps.

The data collection procedure, as depicted in Figure 4(a), was demonstrated using knee X-ray images for osteoarthritis (OA) assessment. The process commenced with radiologists logging into their individual profiles, followed by the calibration of through a standard 5-point calibration routine, as described in previous work (Karessli et al., 2017). Then, the radiologist engaged in a two-step cycle comprising reading and diagnosis to accomplish

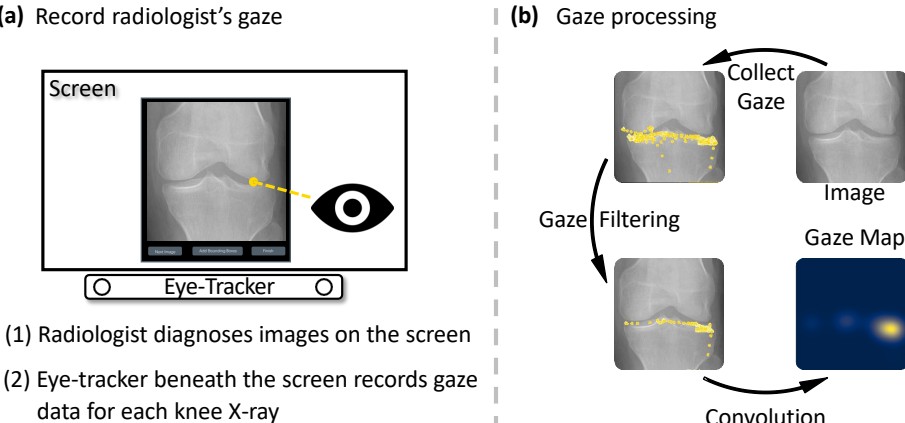

**Figure 4:** (a) Radiologists view knee X-ray images, with their eye movements tracked from image presentation to decision entry. (b) Eye movements are recorded in coordinate-timestamp format, preprocessed to correct distortions, and then transformed into a gaze map (saliency heatmap) used for training the gaze map predictor.

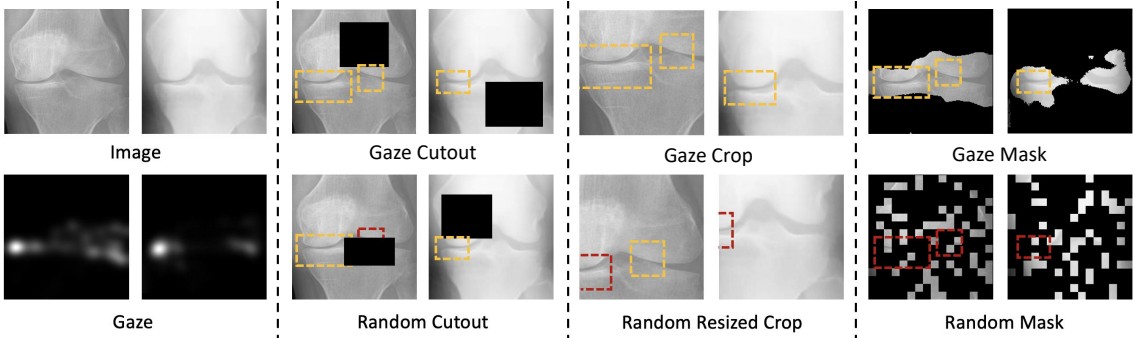

**Figure 5:** On the left, we display an image alongside its corresponding radiologist's gaze map. On the right, we augment the image to create positive pairs. We showcase augmented exemplar images using GCA (top row) and compare them with conventional counterparts (bottom row). Yellow boxes highlight preserved semantic cues (i.e., disease-related abnormal areas) in the augmented images, while red boxes indicate key areas that are absent or partially compromised in conventionally augmented images.

the task of diagnosing knee X-ray images. During the reading phase, an image was randomly selected from our training dataset and presented to the radiologist. The radiologist examined the image until they felt confident in making a diagnosis decision.

In the diagnostic phase, radiologists input their assessments by using number keys on the keyboard, such as "1-4" to denote KL-Grades 1-4 and "Enter" to signify normal or Grade 0. Continuous eye movement data recording takes place throughout both the reading and diagnostic phases. To mitigate radiologist fatigue, we incorporate a 2-minute break for every 20 images. In this study, we collect 354 gaze tracks, which encompass 154 Grade 0 images, 55 Grade 1 images, 81 Grade 2 images, 40 Grade 3 images, and 24 Grade 4 images.

### 3.2. Learning Radiologist's Visual Attention

The training procedure is briefly demonstrated in Fig. 1. As depicted in the figure, we obtain the gaze map based on gaze points for each training image. The knee X-ray image and the gaze map serve as input and output, respectively, to supervise the training of a U-Net (Ronneberger et al., 2015). The trained U-Net, functioning as a gaze map predictor, facilitates the prediction of areas where radiologists are likely to focus their attention when examining knee X-ray images. The U-Net architecture comprises four downsampling encoders and four upsampling decoders, maintaining the same input and output dimensions. Training is conducted over 30 epochs using the Adam optimizer, and the weights from the final epoch are employed in our experimental setup. It is worth noting that minor inaccuracies in the gaze map predictor have minimal impact on subsequent view-filtering processes, as its primary role is to provide an approximate localization of the regions of interest for radiologists.

### 3.3. Crafting Image Views Given Gaze Heatmap

The main idea behind our approach is straightforward: when radiologists identify specific informative areas within an image, these regions should be preserved during the data augmentation process to avoid potential loss of valuable information. Building upon this concept, we introduce three novel augmentation techniques designed to create positive views using our trained gaze map predictor: 1) *Gaze Cutout*, 2) *Gaze Crop*, and 3) *Gaze Mask*. These three augmentations are inspired by the classical image augmentation methods known as *Cutout*, *Random Resized Crop*, and *Random Mask*, respectively. You can refer to Figure 5 for visual examples of how these augmentation techniques are applied.

**Gaze Cutout.** The Cutout technique is a commonly used augmentation method that enhances the robustness and overall performance of neural networks (DeVries and Taylor, 2017; Zhong et al., 2020). It involves randomly masking square regions within the input image, potentially removing important semantic information, such as disease abnormalities. To address this issue, we propose 'Gaze Cutout,' which selectively masks out regions that radiologists are less likely to focus on. The implementation details of 'Gaze Cutout' are provided in Algorithm 1. Specifically, we start by predicting the gaze map for an input image, denoted as $x$. After performing a random cutout, we obtain an augmented view, $v_1$, along with its corresponding gaze map. Subsequently, we calculate the intersection (IOU) between the non-zero regions of the gaze map before and after the random cutout. If both augmented views ($v_1$ and $v_2$) have IOU scores above a predefined threshold (e.g., 0.9), they are preserved as a valid positive pair.

**Gaze Crop.** While *Random Resized Crop* is a widely accepted and effective image augmentation technique in many contrastive learning methods, it inherently lacks awareness of semantic content and may inadvertently remove important details, such as abnormalities in the knee joint. To mitigate this issue, we introduce *Gaze Crop*. Its implementation closely resembles that of *Gaze Cutout*. Zoom-in resizing is only applied when it doesn't remove a substantial portion of regions of interest to radiologists. In *Gaze Crop*, we set the intersection threshold to 0.8.

**Gaze Mask.** The concept of using 'dropout' or 'random mask' (Srivastava et al., 2014) for data augmentation is intuitive and has proven successful in self-supervised learning.

---

**Algorithm 1** Example of Gaze Cutout

---

**Require:** $x$: Input knee X-ray image
**Require:** $f$: Saliency prediction model
  $gaze \leftarrow f(x)$                                        ▷ Predicted gaze map
  **do**
      $v_1, gaze_{v1} \leftarrow \text{random\_cutout}([x, gaze])$
      $v_2, gaze_{v2} \leftarrow \text{random\_cutout}([x, gaze])$
  **while** $\text{IoU}(gaze_{v1}, gaze) > 0.9$ **and** $\text{IoU}(gaze_{v2}, gaze) > 0.9$
      **return** $v_1, v_2$                       ▷ Filtered Positive pair

---

However, while this approach has achieved prominence in natural language processing (Gao et al., 2021), its adoption in computer vision has been relatively recent (He et al., 2021). In our work, we introduce *Gaze Mask*, a technique that masks out areas of an image that are less relevant to the observer. We apply a threshold to the gaze map to mask approximately 80% of less informative regions.

## 4. Experimental Analysis

### 4.1. Setting

**Dataset.** Knee X-ray images used for this study are obtained from the osteoarthritis initiative (OAI), which is a multi-center, longitudinal study targeting OA Nevitt et al. (2006). We followed the data split by Chen et al. (2019), which has 5778 training images and 1656 testing images. Our task is to predict the KL grade which is the most commonly used knee OA severity grading system Kellgren and Lawrence (1957), ranking knee OA severity from Grade 0 (i.e. normal) to Grade 4. In experiments, For our experiments, we assessed the performance of our classification models using various training set sizes: 1% label fraction (55 images, evenly distributed across 5 classes contributing labels), 5% label fraction (288 images), 10% label fraction (577 images), 20% label fraction (1154 images), and 100% label fraction (5778 images). All evaluations were conducted on the 1656 testing images, and we reported the classification accuracy (ACC) alongside the mean absolute error (MAE).

    **Implementation Details.** Following (Chen et al., 2020a; He et al., 2020; Grill et al., 2020), we use ResNet50 as the default backbone. Unless otherwise stated, our models were trained for 200 epochs, with a batch size of 512. We use MMSelfSup (Contributors, 2021) framework to pre-training and adopt the recommended training recipe. During pre-training, we utilized the entire training dataset, comprising all 5778 images.

### 4.2. Experimental results

**Comparisons to handcrafted augmentations.** we conduct a comparative analysis of Gaze Conditioned Augmentation (GCA), a learning-based approach, against commonly employed random augmentation techniques typically used in contrastive pre-training. We also compare GCA with augmentation hyper-parameters that have been exhaustively searched, specifically, an augmentation scale of $1.2\times$ for random resized cropping and a cutout size of

**Table 1:** OAI KL-Grade prediction performance of models trained with different number of labels. ResNet-50 is used as the backbone.

| Method | Label fraction | | | | | | | | | |
|---|---|---|---|---|---|---|---|---|---|---|
| | 1% | | 5% | | 10% | | 20% | | 100% | |
| | ACC↑ | MAE↓ | ACC↑ | MAE↓ | ACC↑ | MAE↓ | ACC↑ | MAE↓ | ACC↑ | MAE↓ |
| From-scratch | 26.03 | 1.370 | 37.62 | 1.216 | 39.79 | 1.163 | 39.25 | 1.189 | 58.10 | 0.614 |
| ImageNet linear | 30.43 | 1.272 | 33.88 | 1.002 | 43.12 | 0.986 | 44.38 | 0.938 | 49.40 | 0.790 |
| ImageNet fine-tune | 33.94 | 1.028 | 38.95 | 1.007 | 55.19 | 0.624 | 58.03 | 0.619 | 63.77 | 0.502 |
| Fine-tune (Random) | 34.84 | 0.980 | 42.61 | 0.958 | 56.52 | 0.606 | 57.55 | 0.624 | 65.70 | 0.445 |
| Fine-tune (Searched) | 36.11 | 0.974 | 44.38 | 0.874 | 58.51 | 0.579 | 60.45 | 0.565 | **67.33** | **0.429** |
| Fine-tune (GCA) | **37.01** | **0.930** | **45.17** | **0.908** | **60.45** | **0.559** | **61.54** | **0.557** | 67.09 | 0.434 |
| Linear-probe (Random) | 32.85 | 1.053 | 36.05 | 1.097 | 50.57 | 0.755 | 51.85 | 0.763 | 55.81 | 0.685 |
| Linear-probe(Searched) | 35.75 | 1.018 | 38.40 | 1.026 | 53.44 | 0.697 | 55.62 | 0.651 | 58.93 | 0.586 |
| Linear-probe (GCA) | **39.07** | **0.961** | **40.76** | **0.872** | **54.71** | 0.658 | **56.39** | **0.649** | **59.66** | **0.583** |

48px. We refer to these two settings as "Random" and "Searched," respectively, in Table 1. It is important to note that for all our comparisons, we utilize the BYOL framework as the standard for contrastive learning.

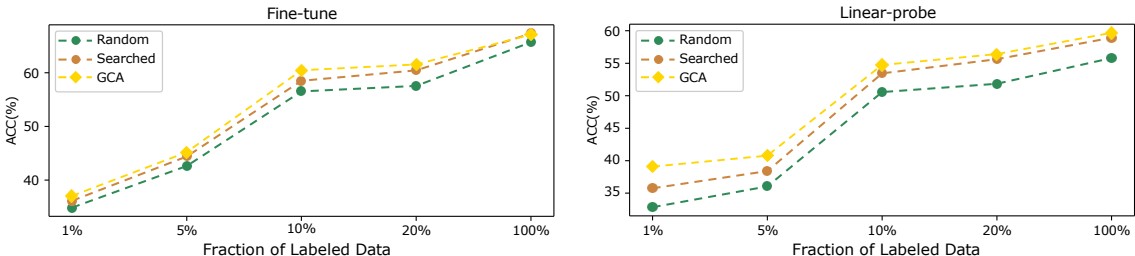

**Figure 6:** Classification accuracy using different augmentation strategies.

In addition to GCA, we include three baseline approaches for comparison in the table: A ResNet-50 network initialized randomly and trained end-to-end (referred to as "From-scratch"). A backbone network, pre-trained on ImageNet, followed by a linear-probing classifier (referred to as "IN linear"). A network pre-trained on ImageNet and subsequently fine-tuned end-to-end (referred to as "IN fine-tune"). We also investigate training efficiency when utilizing classification labels by testing our models on different label fractions of the training data.

From the data presented in Table 1 and Fig. 6, we derive several noteworthy observations: (1) GCA consistently outperforms default contrastive learning in linear-probing performance. GCA (denoted as "Learned," yellow line in Fig.6 bottom panel) consistently surpasses widely adopted default random augmentations (denoted as "Random," green line in Fig.6 bottom panel) by a substantial margin, with an average accuracy gap of +4.90%. These results underscore the capacity of GCA to facilitate the learning of higher-quality representations compared to default random augmentations. (2) In terms of label efficiency, GCA outperforms default augmentation techniques. For instance, linear-probe (GCA) achieves an accuracy of 39.07% at a 1% label fraction, surpassing Linear-probe(Random) at 5% label fraction (accuracy: 36.05%). Moreover, Linear-probe (GCA) at 20% label fraction (accuracy: 56.39%) outperforms Linear-probe(Random) at 100% label fraction (accuracy: 55.81%). (3)

**Table 2:** Ablation study of three proposed augmentation. Linear probing performance on OAI dataset is reported.

| Gaze Crop | Gaze Cutout | Gaze Mask | ACC↑ | MAE↓ |
|:---:|:---:|:---:|:---:|:---:|
| | | | 55.31 | 0.685 |
| | | ✓ | 58.81 | 0.607 |
| ✓ | | | 56.94 | 0.620 |
| | ✓ | | 55.89 | 0.652 |
| ✓ | | ✓ | 56.64 | 0.640 |
| ✓ | ✓ | | 56.64 | 0.636 |
| | ✓ | ✓ | 59.43 | 0.603 |
| ✓ | ✓ | ✓ | 59.66 | 0.583 |

Regarding transferability, GCA (denoted as Fine-tune(GCA)) yields significant improvements compared to the widely adopted approach of pre-training network parameters on ImageNet (denoted as IN fine-tune) and contrastive pre-training with default augmentations (denoted as Fine-tune(Random)). (4) GCA exhibits performance similar to exhaustively searched optimal augmentation settings. In the lower panel of Fig. 6, GCA demonstrates a more substantial advantage in linear-probing performance (compared to searched augmentation, orange line) when fewer labeled data are available. In the upper panel of fine-tune performance, the performance gap is smaller.

In summary, GCA, as a learning-based approach, outperforms widely adopted methods such as ImageNet-based pre-training and contrastive learning with conventional augmentations. It even achieves slightly better overall performance than exhaustively searched optimal augmentation settings. Since collecting gaze data is seamless and effortless compared to the time-consuming and computationally intensive hyperparameter search, GCA presents a practical and valuable solution for developing future computer-aided diagnosis systems.

**Ablation study on the proposed augmentation operators.** To investigate the impact of our three proposed image augmentation methods, we delve into their contributions by evaluating their effectiveness both individually and in combinations during linear probing.

Table 2 presents the linear probing outcomes using a 100% label fraction. Notably, the introduction of "Gaze Mask" augments the linear probing accuracy significantly, elevating it from 55.31% to 58.81%. Moreover, when exclusively implementing "Gaze Crop," we observe a noteworthy boost in linear probing accuracy, reaching 56.94%. Conversely, "Gaze Cutout" yields a comparatively modest improvement in comparison to the other two operators. Recent findings in masked image modeling corroborate these observations (He et al., 2021). This disparity in performance may be attributed to the fact that "Gaze Cutout" only removes a small portion of the image, rendering the contrastive learning task relatively facile when contrasted with the other augmentation techniques.

## 5. Conclusion and Discussion

In this paper, we investigate whether prevalent augmentation techniques used in general-purpose contrastive learning can adversely affect learned representations for medical images. We find that strong augmentations, effective for natural images, can harm pre-training for

medical images due to their vulnerability to corruption, especially in small, semantically important regions. To address this, we propose three learnable augmentation operations guided by radiologists' gaze, using eye-tracking and a visual attention prediction network. Our approach, Gaze Conditiond Augmentation (GCA), significantly improves the performance of network compared to random augmentation, with few lines of extra code.

Our paper demonstrates the viability of learning augmentation strategies from domain experts' visual attention rather than handcrafting them, especially when applying contrastive learning to new domains. In the future, we plan to extend this gaze-guided approach to other self-supervised frameworks, enhancing learning efficiency from unlabeled images.

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
