# OpenReview forum: "Crafting Good Views of Medical Images for Contrastive Learning via Expert-level Visual Attention"
_NeurIPS.cc/2023/Workshop/Gaze_Meets_ML — Gaze Meets ML 2023 Oral_

### Official Review · Reviewer_27XE · 2023-10-15
**Excellent augmentation method for contrastive learning that incorporates gaze into ML**

**Rating:** 9
**Confidence:** 5

**Review:**

Quality

This was a well written paper. Its main points are logical: (1) Some areas within an image are more informative than others. (2) These areas should be preserved when image augmentation is used in training an ML model. (3) These areas can be identified by monitoring gaze of the radiologist.
.
Clarity

The authors record the radiologist's gaze while classifying the stage of arthritis in knee X-rays. They train a U-Net model to predict the radiologist's gaze and create a pixel-wise attention map for gaze.

Three new image augmentation methods are created by modifying the typical cutout, crop, and mask algorithms with this attentional map. Any augmentation views that do not include a significant portion of the gaze-predicted coordinates will be dropped from the positive class pairing in the training of a downstream classification model.

Downstream classification models are trained by contrastive learning. The authors compared classification models using (1) these 3 new augmentation methods, (2) random standard augmentation methods and (3) fine-tuned standard augmentation methods. As expected, classification models that used the gaze-aware augmentation methods scored significantly better on 5%, 10%, 20%, and 100% of the data.

Significance of the Work

This work strongly suggests that image augmentation tools in ML should be based on gaze-informed attentional models. My only criticism is that predicting human gaze may be specific to the image type and model type. It would be a natural extension of the work to determine if a U-Net trained on images from one task could be successfully applied to another. For example, could the gaze U-Net from the knee be used on similar X-rays of arthritis in the finger joints? If not, then a new gaze model would need to be trained. The authors state that their plan to test this in future work. The authors also suggest that this is not a burden, but it does require additional data collection and expense that must be taken into account.

Pros
* Very well written
* The authors followed a straightforward line of reasoning and took the reader through their thought process nicely.
* Figures and tables were excellent.

Cons
* The technique does require an additional U-Net model with gaze data. It is unclear how much of a burden this will be if this technique is applied across the field. Nevertheless, the motivation and results are strong enough to outweigh the additional work.

---

### Official Review · Reviewer_dzpP · 2023-10-21
**Crafting Good Views of Medical Images for Contrastive Learning with eye gaze tracking data**

**Rating:** 8
**Confidence:** 4

**Review:**

This is a well written contribution on using radiologist eye gaze to filter out potentially ‘bad’ positive pairs in contrastive learning. The idea is that random augmentation to generate positive pairs can cause removal or diminishing of disease conditions in medical images. Therefore, the authors build saliency maps from radiologist eye gaze and use those to guide the selection process in positive pair generation. They show good results as improvement in classification performance when compared to random transformations.

Pro: The method is a smart use of eye gaze data. It appears to work. The paper is very easy to read and understand.

Cons: The method is unlikely to generalize well. I assume in knee images, with limited view as used in this work, the focus of the radiologist would be on the knee joint. While the eye gaze map accurately captures that, one could question whether that was not possible to discern without the eye gaze map. In contrast, in a complicated view with many organs (say abdominal images), I’d be curious if the method still works. And I am not sure if we see enough data here to have sense of generalization capability. Nevertheless, I think this is a good first step and recommend acceptance.

---

### Official Review · Reviewer_LzYt · 2023-10-24
**review of using gaze to learn data augmentation**

**Rating:** 7
**Confidence:** 4

**Review:**

The article presents really interesting experiments using gaze tracking in medical images. Such experiments are not easy to conduct and here 354 gaze tracks were recorded. I am not sure but I missed the information on how many were done by each radiologist and how many radiologists in total viewed how many images. This is important to identify imbalances.
Using the gaze information for data augmentation is an interesting approach and in this form seems novel to me. Keeping the region of interest where radiologists focus their gaze unchanged allows algorithms to augment irrelevant parts of the images and thus better identify these parts.
I still feel that augmenting also the region of interest could further improve results, but likely this need to be done in more subtle way to keep the main infirmation intact. Still, ads this part is in clinical imaging subject to modifications and would clearly profit from augmentation. The article of Marini et al. learns data augmentation based on which modifications actually occur in a very large database of histopathology image patches, and something along these lines could possible further improve the strategy of this article.

---

### Meta-Review · Area_Chair_ctUa · 2023-10-26

**Recommendation:** Accept (Oral)
**Confidence:** 4

**Metareview:**

Thank you for this well written and interesting contribution using radiologists gaze for data augmentation in contrastive learning. The authors showed that classification models that used the gaze-aware augmentation methods scored significantly better on 5%, 10%, 20%, and 100% of the data compared to random augmentation. This work suggests that gaze-informed attentional models can be a strong addition to the ML image augmentation toolkit. In the medical domain, this adds invaluable interpretability to the trained models. Overall, recommend accept for oral presentation.

---

### Decision · Program_Chairs · 2023-10-26

Accept (Oral)